# Applications of Innovative Non-Thermal Pulsed Electric Field Technology in Developing Safer and Healthier Fruit Juices

**DOI:** 10.3390/molecules27134031

**Published:** 2022-06-23

**Authors:** Ume Roobab, Afeera Abida, James S. Chacha, Aiman Athar, Ghulam Muhammad Madni, Muhammad Modassar Ali Nawaz Ranjha, Alexandru Vasile Rusu, Xin-An Zeng, Rana Muhammad Aadil, Monica Trif

**Affiliations:** 1School of Food Science and Engineering, South China University of Technology, Guangzhou 510641, China; mahroba73@gmail.com (U.R.); james.chacha@sua.ac.tz (J.S.C.); 2Overseas Expertise Introduction Center for Discipline Innovation of Food Nutrition and Human Health (111 Center), Guangzhou 510640, China; 3National Institute of Food Science and Technology, University of Agriculture, Faisalabad 38000, Pakistan; afeeraabida162@gmail.com (A.A.); aimanathar@gcwuf.edu.pk (A.A.); raimadni0@gmail.com (G.M.M.); 4Department of Food Science and Agroprocessing, School of Engineering and Technology, Sokoine University of Agriculture, Chuo Kikuu, Morogogoro P.O. Box 3006, Tanzania; 5Institute of Food Science and Nutrition, University of Sargodha, Sargodha 40100, Pakistan; modassarranjha@gmail.com; 6Life Science Institute, University of Agricultural Sciences and Veterinary Medicine Cluj-Napoca, 400372 Cluj-Napoca, Romania; 7Faculty of Animal Science and Biotechnology, University of Agricultural Sciences and Veterinary Medicine Cluj-Napoca, 400372 Cluj-Napoca, Romania; 8Department of Food Research, Centre for Innovative Process Engineering (Centiv) GmbH, 28857 Syke, Germany; monica_trif@hotmail.com

**Keywords:** pulsed electric field, non-thermal technology, fresh fruit juice, polyphenol oxidase, peroxidase, pectin methyl esterase, polygalacturonase, health

## Abstract

The deactivation of degrading and pectinolytic enzymes is crucial in the fruit juice industry. In commercial fruit juice production, a variety of approaches are applied to inactivate degradative enzymes. One of the most extensively utilized traditional procedures for improving the general acceptability of juice is thermal heat treatment. The utilization of a non-thermal pulsed electric field (PEF) as a promising technology for retaining the fresh-like qualities of juice by efficiently inactivating enzymes and bacteria will be discussed in this review. Induced structural alteration provides for energy savings, reduced raw material waste, and the development of new products. PEF alters the α-helix conformation and changes the active site of enzymes. Furthermore, PEF-treated juices restore enzymatic activity during storage due to either partial enzyme inactivation or the presence of PEF-resistant isozymes. The increase in activity sites caused by structural changes causes the enzymes to be hyperactivated. PEF pretreatments or their combination with other nonthermal techniques improve enzyme activation. For endogenous enzyme inactivation, a clean-label hurdle technology based on PEF and mild temperature could be utilized instead of harsh heat treatments. Furthermore, by substituting or combining conventional pasteurization with PEF technology for improved preservation of both fruit and vegetable juices, PEF technology has enormous economic potential. PEF treatment has advantages not only in terms of product quality but also in terms of manufacturing. Extending the shelf life simplifies production planning and broadens the product range significantly. Supermarkets can be served from the warehouse by increasing storage stability. As storage stability improves, set-up and cleaning durations decrease, and flexibility increases, with only minor product adjustments required throughout the manufacturing process.

## 1. Introduction

Fruits are vital for the human diet as they contain valuable vitamins and minerals. Although fruits are processed into many forms such as juice, mash, pulp, and canned fruits; their quality deteriorates over time. Fruit juice quality is determined by its enzymatic, physical, organoleptic, and microbiological aspects [1,2,3]. In fruits, quality characteristics such as flavor, texture, color, sensory and nutritional excellence are determined by the key role played by enzymes. The endogenous enzyme activity, for instance, enzymes such as peroxidase (POD), polyphenol oxidase (PPO), pectin methylesterase (PME), polygalacturonase (PG), lipoxygenase (LOX), and β-glucosidase (β-GLUC), affect the quality of fruits post the harvest period. The shelf life of horticultural produce is shortened by the activity of these enzymes together with the growth of microorganisms and/or oxidative reactions. Enzyme inactivation is critical in food processing and preservation [4]. Most of the endogenous enzymes remain active during and after postharvest processing and cause detrimental alterations in the quality characteristics of fruit, for example, color, texture, flavor, and nutrient content [5]. High deteriorating enzymatic activities contribute to the oxidation of polyphenols and enzymatic browning of juices [6]. Thus, efforts need to be undertaken to control enzyme activity in food products with extended shelf life. Traditionally, the degradation due to enzymes has been prevented by a combined approach involving heat treatment, differences in pH, and the utilization of chemical inhibitors. Fruit juices are usually preserved by traditional thermal processing methods which is an economical way to guarantee not only enzymatic inactivation but also microbiological protection [7].

Nonetheless, heat treatment has an unfavorable effect on the final product’s excellence [8,9]. Although heat preserves the juice quality, it is, however, harmful to the sensory, functional, and nutritive features [10,11]. To deactivate the heat-stable isoenzymes, intense thermal treatments are required. This might, however, result in a reduction in the “fresh-like” characteristics of juices because of changes in flavor and odor [12]. There is, therefore, profound attention to the utilization of nonthermal techniques for enzymatic deactivation. The interest in high-quality food products has resulted in the food industry implementing various processing techniques. For this purpose, many conventional thermal processing technologies, such as advanced ohmic, microwave, dielectric, and radio frequency heating, nonthermal hydrostatic pressure (HPP), high-pressure carbon dioxide (HPCD), ultrasound (US), and pulsed electric field (PEF) techniques are being employed in preserving fruit juices [13,14,15,16,17,18,19]. Compared with other traditional thermal techniques, the PEF technique has shown some usefulness including continuous flow, low energy consumption, shorter processing time, and low processing temperature for clean-label processing of fruit juices [20,21,22,23]. PEF uses short pulses of electric field for microseconds to milliseconds at different temperatures (ambient, below or slightly above ambient), and the product is processed by being placed between a set of electrodes [24]. PEF combines electroporation and electro-permeabilization of the cell membrane with an electric field of 10–80 kV/cm applied for a short time (1–100 µs). The number of pulses conveyed to the product generates a low amount of heat, thus preserving taste, flavor, and nutritional components [25].

Electroporation can be permanent, resulting in cell death, depending on the strength of the electric field. Liquid foods are preferable for PEF as current flows more efficiently through liquid food owing to the presence of charged molecules which facilitates the pulse transfer from one point to the other [26]. The nonthermal food preservation impact of PEF has been studied for the enzymatic inactivation, including PPO [27,28,29], POD [11,30], PME [31,32], LOX [11], and PG [33,34]. This study analyzes PEF applications and their combination with other nonthermal technologies for the deactivation of the fruit juices’ endogenous enzymes.

## 2. Working Principle of PEF for Enzyme Inactivation

A PEF system consists of a treatment chamber, a high-voltage power source, a pulse generator, a cooling system for temperature-rise balance during treatment, and an energy discharging switch to electrodes as its main components [32,35]. The generator converts alternate current (AC) to direct current (DC) using a device for charging energy storage devices such as a capacitor. Electrical energy is controlled by a key component that acts as a switch. The treatment chamber (parallel electrodes and collinear tubes) consists of two electrodes, held in place by an insulating substance, forming a food material enclosure. Parallel electrodes consist of a rectangular insulating tube having electrodes on adjacent sides, while the collinear tube consists of an electrically insulating tube with electrodes on both sides. Figure 1 shows a schematic representation of a pulsed electric field process (PEF).

Food conductivity, chamber geometry, circuit parameters, the intensity of the electric field, processing time, frequency, pulse shape, and the specific energy are all factors that influence the efficiency of the PEF process [3]. Electric field strength, which is usually reported in kV/cm, depends on the voltage conveyed as well as the distance between electrodes. The processing (treatment) duration is a function of the number of pulses applied and the pulse width which is reported in “µs”. The specific energy of the pulse depends on food conductivity, the geometry and resistance of the chamber, pulse width, and applied voltage which is reported in “kJ/kg”. Frequency is reported in “hertz” which is pulses per second. Most importantly, the effectiveness of a PEF treatment is determined by the matrix of the fruit and vegetable juice [3]. Moreover, and not to be neglected, especially in critical zones, is the distribution of temperature within the PEF processing chamber. Besides, the pH shift during PEF treatment has been observed and is known to contribute to the partial enzymatic (PPO) deactivation [36].

Reduction in enzyme activity is determined by various parameters, for instance, pulse width, frequency, the strength of the electric field, and treatment time. Pulse polarity (monopolar and bipolar) has appeared as a determining variable for enzyme activity. The monopolar electric field separates the oppositely charged molecules and makes a layer on the electrodes decreasing the efficiency of treatment but in contrast, the bipolar mode avoids this separation and minimizes the deposit on the electrodes [37,38]. However, the effect of pulse width (7 µs) was more pronounced with bipolar PEF to inactivate PG, PPO, and PME in juice made from strawberries [39,40]. Furthermore, 1 µs monopolar pulse achieved the lowest PME (10%) and PG (75%) residual activity (RA) in strawberry juice [41]. These results also indicated that PME was more resistant to PEF than PG as PEF selectively inactivated PG and partially inactivated PME.

Similarly, Aguiló-Aguayo et al. [42] observed the lowest RA of PG (60%), PME (15%), POD (0.16%), and LOX (48.02%) in watermelon juice during processing through bipolar PEF at 250 Hz, 5.5 µs, and 7 µs pulsed width. Results showed that the PEF treatment (35 kV/cm for 1727 µs applying 4 µs pulses at 188 Hz in bipolar mode) led to more than 50% substantial loss of the activity of PME and a slight reduction in the activity of PG [43]. According to the study, low PME activity contributed to the reduction in the substrate for PG action because PG accelerates the hydrolytic splitting of the glycosidic bond in the acid pectin, which formed because of PME de-esterification. However, [44] highlighted the slight PG deactivation is due to the presence of a PEF-resistant PG isoform that carried minor variations in the watermelon juice, especially color. Moreover, the presence of salt bridges and hydrogen bonding had influenced the activity of enzymes during processing. Sørhaug and Stepaniak [45] stated that the higher the hydrogen bonds and salt bridge, the higher the stability or resistance to enzyme inactivation.

Generally, PEF treatment increases the active sites of enzymes [46,47]. Moreover, it changes the secondary bonds which maintain the molecules of enzymes [48]. PEF has been depicted as being able to modify the enzymes’ structure [49,50], thereby influencing the activity of the enzyme, although too high treatment conditions may lead to the damage of the helical structure [51]. Furthermore, the free radicals produced during electrochemical reactions potentially attack enzymes [52]. Interestingly, in some cases, PEF treatments showed no inactivation or even enhancement of enzyme activity. For instance, PEF treatments induced β-GLUC activation in strawberry juice [53]. The suggested optimum temperature for the activation of strawberry β-GLUC was over 60 °C [54]. However, the PEF treatment (below 35 °C) seemed to be the reason for the enzyme’s activation other than high temperature. Similarly, Aguiló-Aguayo et al. [47] noted higher LOX activity after PEF than after thermal treatment at 90 °C for 60 s [53]. However, it is also necessary to keep some RA of LOX because it has a significant role in juice flavor quality. In contrast, a higher decrease in LOX activities was observed during storage in juice samples processed by PEF as compared to those that were thermally processed. Aguiló-Aguayo et al. [41] also proposed that the LOX heat-resistant fraction could also be PEF resistant. However, the commercial LOX solution showed 88.26% deactivation when treated at 24 kV/cm for 962 µs [55]. Generally, the inactivation of enzymes is linked with structural changes in enzymes, which occur in the secondary structure by the forfeiture of α-helix and a β-sheet content upsurge [56].

## 3. Major Applications of PEF in Fruit Juices

In the context of the juice industry, researchers reported beneficial results of PEF for the inactivation of different enzymes in diverse fruit juices including apple [27,30], strawberry [39], watermelon [42,43,57], and citrus juices [31,33,58,59]. Results from several studies indicated that while LOX, PG, and β-GLUC showed fairly greater PEF resistance with less than 50% enzyme activity reduction, PPO, POD, and PME are inclined to PEF treatment with 85–100% deactivation [51]. However, the inactivation rate is determined by PEF treatment conditions, for instance, pulse shape, width and frequency, and treatment chamber geometry, which need to be optimized for obtaining high-quality results. Some of the research results illustrated in Table 1 show that these parameters can modify the inactivation efficiencies for the same commodity. For instance, different combinations of pulse frequency, bipolar pulses (as compared to monopolar mode), and treatment time could be used to drive more positive results for the enzyme inactivation [3].

### 3.1. Effect of Electric Fields on Apple Juice Enzymes

Apple juice (especially unclarified) is gaining an increased market share as one of the most popular fruit juices due to its organoleptic and nutritional characteristics. Unclarified or cloudy apple juice contains more pulp in suspension and has a “fresh like” flavor. Moreover, unclarified apple juice is oxygen-sensitive and has considerable quantities of polyphenols, PPO, and POD. Hence, stringent processing conditions are required to safeguard its superiority, particularly to avert browning due to enzymatic activity without negatively influencing the organoleptic and functional qualities [9]. PPO is an oxidoreductase enzyme containing copper, which results in juice browning and color destruction [68]. During enzymatic browning, melanins and benzoquinone are produced resulting in increased activity of PPO. Its thermal inactivation is also used as an indicator of blanching [69]. Generally, the catalytic activity of PPO can be destroyed under temperatures from 70 °C to 90 °C [70]. On the other hand, POD is an enzyme-containing heme, which is involved in the degradation of pigments and off-flavor development in foodstuffs. It is among the enzymes that are deemed most stable to heat and is commonly utilized as an indicator for the endogenous enzymes’ and microorganism’s inactivation during thermal treatment [71]. PEF, as one of the non-heat treatment techniques, ensures enzyme inactivation without adversely affecting the sensorial as well as the nutritional facets of juices [60]. The effect of PEF on various physicochemical aspects of apples has been broadly studied and resulted in an insignificant effect on color [9,72], pH [73], soluble solids [72], and the composition of vitamin C [74].

A comparative study of thermosonication (1.3 W/mL, 10 min, 58 °C), PEF (24.8 kV/cm, 60 pulses, 169 µs, 53.8 °C), and thermal processing (for 20 min at 75 °C) conducted by [27,30], showed that traditional thermal processing was slightly better than PEF in enzyme inactivation. During storage, apple juice samples that were PEF-processed depicted a decline in the activity of PPO from 17.7% to 11.5–13.5%. However, the thermal treatment caused detrimental impacts on the nutrient composition as well as the quality of apple juice during storage for 30 days at 3 °C and normal temperature [68]. In addition, Wibowoet al. [30] compared HPP, PEF, and thermal processing apple juices and indicated a PPO and POD inactivation exceeding 90%. A 100% inactivation of PME during bipolar PEF treatment (12.3 kV/cm, 2 μs, 132.5 kJ/L, inlet and outlet temperatures of 37.3 °C and 73.8 °C) was observed. According to [75], the reported drop in PME could be ascribed mainly to the heat treatment effects associated with PEF processing that modifies both the tertiary and secondary structures of enzymes causing an activity loss. PEF (preheating to 50 °C combined with 100 μs at 40 kV/cm) caused a 71% and 68% reduction in the activity of POD and PPO enzymes, respectively in apple juices [76]. Several findings have shown a positive linear relationship between the processing factors (i.e., treatment time, the intensity of the electric field strength, pulse width) as well as the reduction of PPO and POD in juices made from apple [60,76], grape juice [62] and buffer solution [77]. In research by Bi et al. [60], the RA of PPO and POD was reduced by 7.1–98.5% and 9.6–94.2%, respectively, as compared to control samples while an upsurge in pulse time of 2 µs caused the greatest inactivation for both enzymes at 35 kV/cm. Furthermore, a decrease in RA of POD and PPO in relation to the increase in the intensity of PEF was observed [60]. According to Luo et al. [78] and Zhong et al. [77], the denaturation of enzymes might be a viable reason for the linear decrease in enzymatic activity. However, the pulse rise time of 2 µs depicted had a higher deactivation of both enzymes due to a higher rise in samples’ temperatures (9.2–20.7 °C) and more energy density input than the pulse rise time of 0.2 µs at a similar intensity of the electric field [60]. The temperature rise contributed to some thermal inactivation of both enzymes, while the activity of the enzyme decreased with the increase in the input of energy density, which was determined by the design of the treatment chamber, the intensity of the electric field, the processing duration and the conductivity of the product [9,56].

Riener et al. [76] perceived that an increase in the pre-processing temperature of the juice had a substantial influence on the PPO and POD inactivation. For instance, the RA of POD and PPO declined to 45% and 43%, respectively after preheating the juice to 50 °C followed by PEF treatment at 30 kV/cm for 100 µs. With a pre-processing temperature of 23 °C, the RA of juice was 63.6% for POD and 59.4% for PPO in similar PEF conditions. It was further established that PEF processing at 30 kV/cm resulted in a modest increase in juice temperature in the PEF cell. Irrespective of the inlet temperature, the increase remained unchanged at about 15 °C. The highest temperature attained by the juice was 65 °C under these explicit PEF settings. Moreover, the maximum electric field intensity (40 kV/cm) resulted in an RA of 28.9%, although as stated earlier, this field intensity also raised the temperature of the juice from 50 °C to 72 °C approximately. This temperature is near to the one used in minimal juice pasteurization and could therefore contribute to some heat deactivation of both PPO and POD. However, a drawback of the intensity of the electric field to 30 kV/cm is prudent to avoid the detrimental effects of thermal treatment on juice excellence as, under these circumstances, the outlet temperature of the juice would not surpass 65 °C. Similarly, extending the processing time from 25 to 100 µs linearly declined the RA for both PPO and POD, from 54% to 75%, and from 48% to 67%, respectively [76].

Synergistic effects of PEF (30 kV/cm) and heat (preheated to 40 °C) were also observed for apple juices with 48% PPO inactivation (complete inactivation at 60 °C) and undetectable levels of 5-hydroxymethylfurfural [71]. Similarly, [79] reported substantially low RA of POD and PPO in PEF-treated apple juice (10–30 kV/cm for 200–1000 µs at 20–60 °C), with pasteurization at 90 °C for 5 min found to be more effective. In addition, for both enzymes, with complete deactivation at 30 kV/cm, 1000 µs, and 60 °C, fewer color modifications in juices as compared to thermal pasteurization were reported. The highest carrot and apple juice POD and PPO inactivation that was preheated to 80 °C was achieved by [68]. The high temperature resulted in a rise in the enzymatic internal energy, thereby causing the breaking of the bonds responsible for the enzymatic three-dimensional structure. The PEF treatment (33–42 kV/cm with frequencies of 150–300 pulses/s) was compared with conventional ultra-high temperature (UHT) pasteurization (115–135 °C for 3–5 s) regarding PPO inactivation in apple juices by [80]. The UHT treatment attained a 95% decrease in the activity of PPO as compared to PEF treatment which reduced 70% of PPO (at 38.5 kV/cm and 300 pulses/s at 50 °C). However, the conventional UHT pasteurization triggered biochemical reactions that subsequently resulted in quality changes in pH, color, soluble solids, and acidity. Tian et al. [81] reported that the combined effect of radiofrequency (RF; electromagnetic waves of low frequency) and PEF had improved the quality of apple juice. Preprocessing of apple tissues with 5–10 min RF treatment (27.12 MHz) decreased the PPO activity to 67–86%. After RF application, the juice was squeezed and PEF-treated at 15–35 kV/cm for 400 µs which further decreased the enzyme activities.

### 3.2. Effect of Electric Fields on Citrus Juice Enzymes

Fresh orange juice, which is rich in vitamin C content, is among the most popular and consumed juices globally. PME is the major enzyme in fresh orange juices related to adverse quality losses in terms of juice clarification or gelation. The juice cloud comprises finely divided particles of cellulose, pectin, hemicellulose, proteins, and lipids in suspension. However, it shows a loss of cloudiness and concentrates gelation a short time after squeezing, due to the PME activity. As a cell wall-bound enzyme, PME is found in all citrus fruits formulating a complex with pectin via electrostatic interactions. During the extraction process, the enzyme is released into the juice hydrolyzing the pectin (methyl esters of homogalacturonan) and changing it progressively to low methoxy pectin and pectic acids, which form calcium ions insoluble complexes, leading to the loss of the cloud and pectin precipitation. PME is responsible for the hydrolysis of pectin, which results in the instability of the cloud as well as the decrease in viscosity by the degradation of the pectin chain [82]. Cloud stability plays a key role in the quality characteristics of citrus juices such as turbidity, flavor, aroma, and color. PG and PME are linked with the de-esterification of pectin and hydrolytic cleavage of the α-1,4-glycosidic bonds. PME acts on pectin by cleaving methyl esters to render poly-D-galacturonic acid, which modifies the appearance of citrus juices. If these enzymes are not inactivated, they ultimately destroy the citrus fruit juices’ cloudy stability. Hence, enzyme inactivation is necessary to avoid quality losses in fresh orange juices.

In citrus juices, PME occurs in different isoforms (20–33% heat-stable fraction), with different heat stabilities, with respect to the fruit type [83]. Generally, heat-stable fractions of PME need stark thermal processing for comprehensive deactivation as compared to thermal-labile forms of the enzyme, which need modest temperatures (50–60 °C) for deactivation [83]. In oranges from Valencia possessing 5% thermostable PME, it has been revealed that only temperatures above 72 °C can inactivate this portion, which can deteriorate the functional and nutritional quality of the juice [84]. Besides, a substantial electric field, and both extensive treatment durations and greater preheating temperatures, resulted in a more extensive degree of inactivation of PME. While compared with the other processing techniques such as HPP and thermal processing, PEF treatments were unsuccessful in terms of orange juice pasteurization. Vervoort et al. [59] reported 34% inactivation of PME and POD using PEF at 23 kV/cm, monopolar pulses of 2 µs duration, and 38 °C. However, the other quality parameters of orange juice regarding public health such as organic acids, sugars, vitamin C, bitter compounds, carotenoids, furfural, and 5-hydroxymethylfurfural were unaffected by the processing. Depending on the enzyme properties and experimental circumstances, it may be argued that both PEF and the heat created can lead in perceived deactivation. Therefore, it is important to design PEF treatment equipment in such way that a sensible rise in the juice temperature is produced during its passage through the cell. Such a rise is reliant on both the pulse number and the electric field strength.

The effect of various electric field strengths combined with non-lethal processing temperatures (<50 °C) to inactivate orange PME has been studied whereby 80% inactivation at 35 kV cm^−1^ for 1500 µs and 37.5 °C [66], and 90% inactivation at 35 kV/cm for 59 µs and 25.1 °C initial temperature has been reported [64] as compared to pasteurization at 90–95 °C for 30 s^−1^ min. Maximum PME inactivation of 81.4% and 81% using PEF at 25 kV/cm for 340 µs and 330 µs with 70 and 63 °C respectively were observed in orange-carrot juices [44,85]. Moreover, the combined effect of PEF treatments (40 kV/cm for 100 µs) and heat (preheating to 50 °C) enhanced the inactivation rate of PME to 96.8% in red grapefruit juice [83]. Espachs-Barroso [86] stated that the thermal treatment from 54 °C to 81 °C for 120 min treatment time could successfully inactivate PME in orange juice. However, 87% PME inactivation was observed in oranges at 19.1 kV/cm for 1.6 ms at 0.5 or 5 Hz pulse frequency [86].

PME can precipitate the juice solids during storage, if not inactivated. An inactivation kinetic model was established to assess the effectiveness of PEF processing on the activity of PME in orange juices during storage (at 4 °C for 180 days) and compared with heat pasteurization at 90 °C for 20 s [31]. The authors observed 93.8% enzyme inactivation during PEF (at 25.3 kV/cm for 1206.2 µs) as compared to 95.2% PME inactivation during heat pasteurization. However, the PME activity of PEF-processed orange juices continued reducing during storage while the heat-processed samples revealed enzyme restoration during storage. At the end of storage, the reduction in PME activity was greater under low-intensity PEF (13 kV/cm for 1033 µs) than under high-intensity PEF (25.3 kV/cm for 1206.2 µs), which might be due to the sub-structural changes in the PME molecules at low intensity of the electric field. The conformational changes in the α-helix of enzymes led to the activity loss during treatment because the α-helix relative content declined after the treatment. These molecules had shown more reduction during storage due to incomplete inactivation and could not restore themselves, thus displaying a substantial decline. Although the exact mechanism to elucidate the greater decline is unknown, it is also assumed that PME reduction might be due to incomplete inactivation of microorganisms, which could utilize the residual PME as a source of nitrogen for growth for the period of storage. At the start of PEF treatment, protein molecules polarize and interact with hydrophobic or covalent bonds that generate aggregates [87]. The enzyme’s active sites are ultimately modified which makes it more problematic for the substrates to draw together, ultimately decreasing the values of the residual activity [88]. High-intensity PEF, on the other hand, resulted in greater microbial inactivation and enzymatic deactivation than damage, resulting in a lesser decrease in PME activity. [31]. With respect to Aguiló-Aguayo [43], the activities of the enzyme in the untreated juice also declined for the storage period of the first two weeks as an impact of the fast growth of the spoilage microbes within the sample. Thus, the treated juices did not display a dramatic enzymatic activity reduction due to the result of the stability of microbes attained with the treatments.

Some nonthermal combinations have also been studied for orange juice PME inactivation. For instance, subjected orange juice to thermosonication and thereafter PEF (40 kV/cm, 55 °C for 10 min) and noted 12.8% minimum and 82.7% maximum RA of PME for 150 µs and 50 µs, respectively [89]. Moreover, Caminiti et al. [58] combined PEF (24 kV/cm, 18 Hz, 93 µs) and mano-thermosonication (400 kPa, 35 °C, 1000 W, 20 kHz) to inactivate PME (19% RA) while the individual treatments, i.e., PEF and manothermosonication reported 86% and 23% RA of PME after treatment. 

### 3.3. Effect of Electric Fields on Miscellaneous Juices Enzymes

Untreated mango juice undergoes enzymatic and microbial degradation. PEF of 35 kV/cm for 1800 μs resulted in 70%, 69.9%, and 46% reduction in PPO, LOX, and POD activities, respectively, immediately after treatment [11]. While during the storage (16–75 days), PPO and POD RA continued to reduce notably. Additionally, the RA of POD was lowest (17.4%) at 49 days owing to the susceptibility of the α-helix structure of POD [90]. In contrast, LOX required more time to reduce 50% from the initial activity to the final storage period. PEF weakened the affinity of enzymes to make a complex with the substrate and aggregates formation due to polarization and interactions in protein molecules that change the active sites of enzymes [33,78].

PEF processing retained higher aroma-related enzyme activities and reduced the major volatile compounds (methyl butanoate ethyl butanoate and linalool) ultimately improving the flavor quality of strawberry juice for up to 14 days [53]. The maximal RA of β-GLUC was achieved at 35 kV/cm for 1700 μs using pulses of 4 μs at 100 Hz in bipolar mode than with heat treatments at different temperatures. Furthermore, Aguiló-Aguayo [40] observed less HMF content, greater luminosity, redness, and accumulation of brown pigments, and improved viscosity in strawberry juices processed using PEF than those processed thermally. PEF application on berry fruits (e.g., red raspberry and blueberry) revealed its minimal impact on PPO activity. The RA of PPO in raspberry and blueberry was 98 and 80%, respectively after 25 kV PEF application [28].

PEF of 25 kV in combination with ultrasound treatment of 24 kHz significantly reduced the activity of PPO in both blueberry and raspberry [28]. According to Noci et al. [61], PEF combined with ultrasound showed comparatively more RA of PPO and POD than thermal treatment. Moreover, the combined influence of PEF and ultrasound can be applied in an independent sequence of technologies—either PEF before US or US before PEF. Table 2 summarizes the variety of factors utilized in PEF in combination with other technologies to successfully inactivate various enzymes representative in apple and citrus juices. However, the combination with mild temperatures reduces the resistance of molecules and increases the ion conductivity of samples [91], which is significant for the enzyme inactivation [92].

## 4. Conclusions

PEF can be used as an innovative technology in various areas of the food industry and bioprocess engineering. The PEF technique for food preservation and a better quality end-product has been continually implemented from the lowest levels (laboratory and pilot) to the industrial levels, especially for liquid food. The benefits of PEF treatment are not only related to product quality but also manufacturing. Extending the shelf life simplifies production planning and greatly expands the product range. Because of the extended shelf life, supermarkets can be served from the warehouse. Another advantage is that by increasing storage stability, fewer product changes are required within the production, reducing set-up and cleaning times and thus increasing flexibility. PEF has been found capable of inactivating the quality deteriorated enzymes for better preservation of fruit juices. Enzyme structure was found labile and sensitive to electric pulses and effectively inactivated at low temperatures. As a non-thermal technology, PEF is a sustainable approach toward better fruit juice production with enhanced color, flavor, nutrition, safety, and stability of juices. PEF-processing parameters can easily be optimized, either to obtain the desired level of inactivation or to increase enzyme activity according to the process requirement. PEF treatment means that sensitive flavors and nutrients are better preserved, allowing for higher product quality. PEF technology is acquiring considerable attention towards its influence as an efficient nonthermal, clean and green preservation technology while keeping fresh-like properties in the food industry sector. Although findings have been reported on PEF-mediated deactivation, enzymes’ stability parameters, as well as the impacts of storage on enzymatic activity during shelf life, the related studies are scanty, warranting further studies in the future.

## Figures and Tables

**Figure 1 molecules-27-04031-f001:**
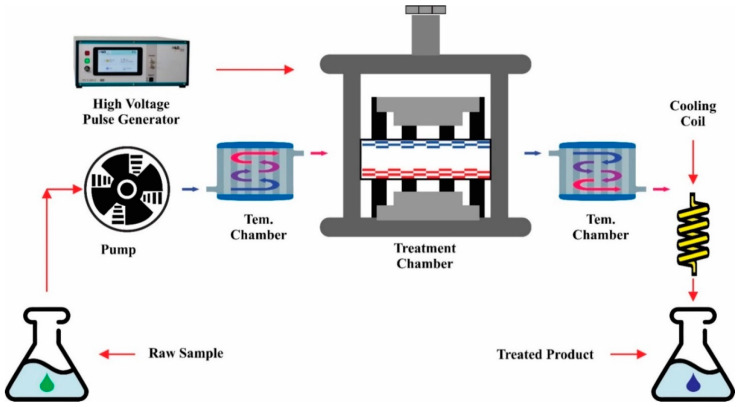
A representation of a pulsed electric field process (PEF) (Adapted from Ranjha et al. [20]).

**Table 1 molecules-27-04031-t001:** Pulsed Electric Field Effects on the Enzymatic Profile of Juices.

Sample	Target Enzyme	Experimental Design			Compared with	Effect	Ref.
		Electric Field Strength + Time	Frequency	Pulse Width			
Strawberry juice (cv. Camarosa)	PG, PME, LOX, and β-GLUC	35 kV/cm for 1700 µs.	100 Hz	4 µs, bipolar mode.	TP: 90 °C for 60 s and 30 s.	73%, 10%, and 66.7%, RA of PG, PME, and LOX, respectively. 15.6% increase in β-GLUC activity.	[40,47,53]
Strawberry juice (cv. Camarosa)	PPO	35 kV/cm for 1000–2000 µs.	50–250 Hz	1.0–7.0 µs, monopolar or bipolar mode.	N.A.I	RA of PPO reduced by 2.5%.	[39]
Apple juice (*Malus domestica* Fuji)	PPO and POD	0–35 kV/cm. Pulse rise time (0.2–2 µs).				RA of PPO and POD was 7.1–98.5% and 9.6–94.2%, respectively.	[60]
Red raspberry (*Rubus strigosus*) and blueberry (*Vaccinium corymbosum*)	PPO	25 kV for 66 µs.	600 Hz	N.A.I	US: 24 kHz, 400 W, 20 min. PEF+US: 600 Hz, 25 kV for 66 μs + 24 kHz for 20 min.	98 and 80% RA of raspberry and blueberry purees were observed respectively.	[28]
Apple juice	POD and PPO	40 kV/cm for 1–100 µs.		N.A.I	UV: 254 nm, 30 W for 30 min.	42% and 47.5% RA of PPO and POD, respectively.	[61]
Apple (*M. domestica* cv. Royal Gala)	PPO	PEF: 24.8 kV/cm for 169 ms, pulses 60, 53.8 °C. Storage: 3 °C and 20 °C for 30 days.	N.A.I	N.A.I	TP: 75 °C, 20 min. TS: 1.3 W/mL for 10 min, 58 °C.	17.7% RA of PPO decreased to 13.5% and 11.5% during storage at 3 and 20 °C, respectively.	[27]
Cloudy apple juice (Belgian apple cultivars)	PPO, POD, and PME	12.5 kV/cm, 27.6 L/h, T_inlet_ 37.6 °C, T_outlet_ 59.5 °C.	62 Hz	N.A.I	TP: 72 °C for 15 s and 85 °C for 30 s. HPP: 400 MPa for 3 min, 600 MPa for 3 min.	36%, 49%, and 50% reduction in PPO, POD, and PME activity.	[30]
		12.3 kV/cm, 24.5 L/h, T_inlet_ 37.3 °C, T_outlet_ 72.8–73.8 °C.	94 Hz	N.A.I	TP: 72 °C for 15 s and 85 °C for 30 s. HPP: 400–600 MPa for 3 min.	>90% PPO and POD inactivation and no PME activity.	
Grape juice (*Vitis vinifera* cv. Parellada)	PPO and POD	25–35 kV/cm for 1–5 µs.	200–1000 Hz			100% PPO and 50% POD inactivation.	[62]
Mango juice (*M. indica* L.) cv. *tommy atkins*	PPO, POD, LOX	35 kV/cm for 50–2000 µs. Storage, 4 °C for 75 days.	200 Hz	N.A.I	TP: 90 °C for 60 s.	70%, 53%, and 44%, PPO, LOX, and POD RA respectively in 1800 µs.	[11]
Orange-carrot juice	PME	24 kV/cm for 93 µs.	18 Hz	N.A.I	TP: 72 °C for 3.5 min. UV: 10.62 J/cm ^2^ for 1 min. HILP: 3 Hz, 3.3 J/cm ^2^ for 360 µs, 30 °C. MTS: 20 kHz, 1000 W, 400 kPa for 2.2 min, 35 °C.	86% RA of PME.	[58]
Orange juice (Kozan-specific variety)	PME	13.8-25.3 kV/cm for 1033–1206 µs;	500 Hz	N.A.I	TP: 90 °C for 10 s and 20 s.	93.8% enzyme inactivation at 25.26 kV/cm–1206.2 µs.	[31]
Orange juice (Valencia oranges)	PME	0–35kV/cm for 184 and 250 ms at 10–50 °C.	N.A.I	N.A.I	TP: 10–50 °C.	90% enzyme inactivation at 25 kV/cm at 50 °C.	[63]
Orange juice (Valencia oranges)	PME	35 kV/cm for 59 µs.		N.A.I	TP: 94.6 °C for 30 s.	88% enzyme inactivation.	[64]
Orange juice (Navelina oranges)	PME and POD	35 kV/cm for 1000 µs.	200 Hz	4 μs pulses in bipolar mode.	TP: 90 °C for 1 min.	81.6% and 100% inactivation of PME and POD, respectively.	[65] [66]
Orange juice	PME and POD	23 kV/cm	90 Hz	2 μs pulses in a monopolar mode.	TP: 72 °C for 20 s.	60.7% and 68.4% RA of PME and POD, respectively.	[59]
Orange juice (Navelina oranges)	POD	5–35 kV/cm for 1500 µs at <40 °C.	50–450 Hz	Pulse width (1–10 µs) in mono and bipolar mode.	TP: 90 °C for 1 min.	5% RA of POD at monopolar and 7% at bipolar. The monopolar mode was more effective.	[67]
Watermelon juice (*Citrullus lanatus* cv. Sugar Baby)	POD, LOX, PME, and PG	35 kV/cm for 1727 µs. Storage, 56 days.	188 Hz	4 µs pulses in bipolar mode.	TP: 90 °C for 30 s and 60 s.	1.7%, 85%, 34.8% and 86.4% RA of POD, LOX, PME and PG.	[43]
Watermelon juice (*Citrullus lanatus* cv. Sugar Baby)	POD, LOX, PME, and PG	35 kV/cm for 1000 µs.	50–250 Hz	Pulse width (1.0–7.0 µs) in monopolar or bipolar mode.	N.A.I	0.16%, 48.02%, 15%, and 60% RA of POD, LOX, PME, and PG.	[42,57]
Fruit juices blend (orange, kiwi, mango, and pineapple)	PME and PG	35 kV/cm	200 Hz	4 μs pulses in bipolar mode.	TP: 90 °C for 1 min.	58.77%, and 73.08% RA of PME and PG, respectively.	[33]

RA: Residual activity; PEF: Pulsed electric field; TP: Thermal processing; US: Ultrasonication; UV: Ultraviolet; HPP: High-pressure processing; HILP: High-intensity light pulses; MTS: Manothermosonication; PME: Pectin methyl esterase; PPO: Polyphenol oxidase; POD: Peroxidase; PG: Polygalacturonase; LOX: Lipoxygenase; β-GLUC: β-glucosidase; N.A.I: Non-available information.

**Table 2 molecules-27-04031-t002:** Combined PEF Treatments with Other Technologies.

Sample	Target Enzyme	Treatment	Experimental Design	Effect	Ref.
Red raspberry (*R. strigosus*) and blueberry (*V. corymbosum*)	PPO	PEF+US	PEF: 600 Hz, 25 KV for 66 µs. PEF+US: 600 Hz, 25 kV for 66 µs, 24 kHz for 20 min; US, 24 kHz, 400 W, 20 min.	Significant (*p* < 0.01) reduction of PPO activity in both raspberry and blueberry.	[28]
Orange juice	PME	PEF+TS	PEF: 30 kV/cm for 25–150 µs, 55 °C for 10 min. HTST: 94 °C for 26 s.	RA decreases 86.5 to 43.2%.	[89]
			PEF: 40 kV/cm for 25–150 µs, 55 °C for 10 min. HTST, 94 °C for 26 s.	RA decreases 82.7 to 12.8%.	
Orange-carrot juice	PME	PEF+MTS	PEF: 24 kV/cm, 18 Hz, 93 µs. MTS: US, 560 W, 5 min; H, 40 °C; HPP, 350 MPa.	19% PME RA.	[58]
Orange-carrot juice	PME	PEF+H	PEF: 767–904 Hz 25 kV/cm, 280–330 µs, 112–132 pulses. H: 68, 70 °C. HTST: 98 °C for 21 s.	75.6–81.4% enzyme inactivation	[85]
Orange-carrot juice	PME	PEF+H	PEF: 25–40 kV/cm, 0–340 µs. H: 63 °C.	81.4% enzyme inactivation.	[44]
Orange juice, milk-based beverage	PME	PEF+H	PEF: 15–30 kV/cm, 25–65 °C H: 60 to 90 °C for 1 min.	At 25 °C increase in PME activity was between 11 and 60%. At 65 °C (30 kV/cm), 91% inactivation. At 80 °C (3–5 kV/cm, 3000–3500 Hz, 1 μs) <10%. PME inactivation.	[93]
Apple juice	PPO	PEF+H	PEF: 33–42 kV/cm, 150–300 pulses/s H: 50 °C. UHT: 115, 125, and 135 °C for 3 and 5 s.	70% reduction of RA at 38.5 kV/cm.	[80]
Apple juice	POD and PPO	PEF+H	PEF: 20–40 kV/cm for 25–100 µs. H: 25, 35, and 50 °C. CP: 72 °C for 26 s.	71% and 68% highest decrease in the enzymatic activity of PPO and POD, respectively.	[76]
Apple juice cv. (Braeburn)	POD and PPO	PEF+H	PEF: 15–35 kV/cm, pulse width (3 to 8 µs). H: 60 °C.	79.8 to 0% and 92 to 6.9% RA of POD and PPO, respectively.	[71]
Apple juice (*Malus pumila* Niedzwetzkyana Dieck)	POD and PPO	PEF+H	PEF: 10–30 kV/cm for 200–1000 µs, 20–60 °C. H: 80, 90, and 115 °C for 10 min, 5 min, and 5 s.	0.04% and 0.16% RA of POD and PPO at 30 kV/cm for 1000 µs and 60 °C, respectively.	[79]
Apple juice	POD and PPO	UV+PEF	PEF: 40 kV/cm for 100 µs. UV: 254 nm, 30 W for 30 min. 40 kV/cm for 1 µs.	47.2% and 42.8% RA of POD and PPO, respectively.	[61]
		PEF+UV		49.5%, and 41.3% RA of POD and PPO, respectively.	
Apple juice (*M. domestica Borkh*. cv. Red Fuji)	PPO	PEF+RF	PEF: 15–35 kV/cm for 400 µs. RF: 27.12 MHz 3.5 kW, 35 mm pole space. H: 60–70 °C for 10 min.	13.57% RA after 10 min preprocessing, 5% RA, 15 kV/cm for 400 µs increase lightness and maintain flavor.	[81]

PME: Pectin methyl esterase, US: Ultrasound; H: Heat; RF: Radio frequency; MS: Manosonication; TS: Thermosonication; MTS: Manothermosonication; UV: Ultraviolet; HILP: High-intensity light pulses; CP: Conventional pasteurization; HTST: High treatment short time; PPO: Polyphenol oxidase; POD: Peroxidase.

## Data Availability

Not applicable.

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
