# Peer review of "Applications of Innovative Non-Thermal Pulsed Electric Field Technology in Developing Safer and Healthier Fruit Juices"

_molecules, 2022, doi:10.3390/molecules27134031_

Round 1
Reviewer 1 Report
Minor concerns:
Comments to the Author:
The manuscript's title is appropriate.
Abstract: The Background of the abstract is well written. The main procedure and findings of the study are well expressed.
Introduction: A brief survey of existing literature, purpose, importance, and innovation of the research is well mentioned.
Tables and graphs are well prepared.
Point 1: In the last sentence of the abstract, it is recommended to give a suggestion for the future.
Point 2: In the introduction, it is recommended to mention the advantages of non-thermal technologies compared to thermal technology. At the same time, write down the types of non-thermal technologies and different areas of use. I recommend the following articles (Cano-Lamadrid and Artés-Hernández 2021; Chiozzi, Agriopoulou, and Varzakas 2022; Yıkmış 2019; Yıkmış et al. 2021)
Cano-Lamadrid, Marina, and Francisco Artés-Hernández. 2021. “By-Products Revalorization with Non-Thermal Treatments to Enhance Phytochemical Compounds of Fruit and Vegetables Derived Products: A Review.” Foods 2022, Vol. 11, Page 59 11(1):59. doi: 10.3390/FOODS11010059.
Chiozzi, Viola, Sofia Agriopoulou, and Theodoros Varzakas. 2022. “Advances, Applications, and Comparison of Thermal (Pasteurization, Sterilization, and Aseptic Packaging) against Non-Thermal (Ultrasounds, UV Radiation, Ozonation, High Hydrostatic Pressure) Technologies in Food Processing.” Applied Sciences 2022, Vol. 12, Page 2202 12(4):2202. doi: 10.3390/APP12042202.
Yıkmış, Seydi. 2019. “Optimization of Uruset Apple Vinegar Production Using Response Surface Methodology for the Enhanced Extraction of Bioactive Substances.” Foods 8(3):107. doi: 10.3390/foods8030107.
Yıkmış, Seydi, Filiz Aksu, Sema Sandıkçı Altunatmaz, and Başak Gökçe Çöl. 2021. “Ultrasound Processing of Vinegar: Modelling the Impact on Bioactives and Other Quality Factors.” Foods 10(8):1703. doi: 10.3390/FOODS10081703.
Point 3: The conclusion part can be shortened a little
Point 4: If possible, a graphic describing the mechanism of action of PEF technology on the cell can be added to the study.
Author Response
Comments and Suggestions for Authors
Minor concerns:
Comments to the Author:
The manuscript's title is appropriate.
Abstract: The Background of the abstract is well written. The main procedure and findings of the study are well expressed.
Introduction: A brief survey of existing literature, purpose, importance, and innovation of the research is well mentioned.
Tables and graphs are well prepared.
Reply: We’re really very thankful to reviewer for their valuable comments in terms of improving our manuscript.
Point 1: In the last sentence of the abstract, it is recommended to give a suggestion for the future.
Reply: Dear reviewer, the future suggestions/recommendations are usually added in the conclusion part and at the end of conclusion; suggestions for future are added (highlighted in yellow). Hope you will agree with this approach. Thank you.
Point 2: In the introduction, it is recommended to mention the advantages of non-thermal technologies compared to thermal technology. At the same time, write down the types of non-thermal technologies and different areas of use. I recommend the following articles (Cano-Lamadrid and Artés-Hernández 2021; Chiozzi, Agriopoulou, and Varzakas 2022; Yıkmış 2019; Yıkmış et al. 2021)
Cano-Lamadrid, Marina, and Francisco Artés-Hernández. 2021. “By-Products Revalorization with Non-Thermal Treatments to Enhance Phytochemical Compounds of Fruit and Vegetables Derived Products: A Review.” Foods 2022, Vol. 11, Page 59 11(1):59. doi: 10.3390/FOODS11010059.
Chiozzi, Viola, Sofia Agriopoulou, and Theodoros Varzakas. 2022. “Advances, Applications, and Comparison of Thermal (Pasteurization, Sterilization, and Aseptic Packaging) against Non-Thermal (Ultrasounds, UV Radiation, Ozonation, High Hydrostatic Pressure) Technologies in Food Processing.” Applied Sciences 2022, Vol. 12, Page 2202 12(4):2202. doi: 10.3390/APP12042202.
Yıkmış, Seydi. 2019. “Optimization of Uruset Apple Vinegar Production Using Response Surface Methodology for the Enhanced Extraction of Bioactive Substances.” Foods 8(3):107. doi: 10.3390/foods8030107.
Yıkmış, Seydi, Filiz Aksu, Sema Sandıkçı Altunatmaz, and Başak Gökçe Çöl. 2021. “Ultrasound Processing of Vinegar: Modelling the Impact on Bioactives and Other Quality Factors.” Foods 10(8):1703. doi: 10.3390/FOODS10081703.
Reply: Dear reviewer, in the introduction section a detailed paragraph have been already added to discuss the advantages of non-thermal technologies (highlighted in yellow). Also, out of your suggested references, most relevant studies have been added to support the discussion. Also, following studies have been incorporated in the said paragraph to support the discussion.
Yıkmış, S. Sensory, physicochemical, microbiological and bioactive properties of red watermelon juice and yellow watermelon juice after ultrasound treatment. Food Measure 14, 1417–1426 (2020). https://doi.org/10.1007/s11694-020-00391-7
Yıkmış, S., Bozgeyik, E. & Şimşek, M.A. Ultrasound processing of verjuice (unripe grape juice) vinegar: effect on bioactive compounds, sensory properties, microbiological quality and anticarcinogenic activity. J Food Sci Technol 57, 3445–3456 (2020). https://doi.org/10.1007/s13197-020-04379-5
Point 3: The conclusion part can be shortened a little
Reply: Dear reviewer, thank you so much for your valuable suggestion but it is requested that shorting the conclusion part is not recommended as the most appropriate cessations have been added there.
Point 4: If possible, a graphic describing the mechanism of action of PEF technology on the cell can be added to the study.
Reply: Dear reviewer, thank you so much for your valuable suggestion. During the preparation of early drafts of this review, auuthors mutually discussed about such mechanism and also literature was assessed for this. But then it was mutually agreed by the authors that such type of mechanism may only be added to illustrate the effect of PEF on extraction of plant material. As it has been added in:
Ranjha, M.M.A.N.; Kanwal, R.; Shafique, B.; Arshad, R.N.; Irfan, S.; Kieliszek, M.; Kowalczewski, P.Ł.; Irfan, M.; Khalid, M.Z.; Roobab, U.; et al. A Critical Review on Pulsed Electric Field: A Novel Technology for the Extraction of Phytoconstituents. Mol. 2021, 26.
Reviewer 2 Report
Comments to author:
The work described in the present manuscript is consistent with the scope of the journal.
Authors described the utilization of non-thermal pulsed electric field (PEF) as a promising technology to produce juices with improved properties. The review is very well organized by the type of fruit juice and extensively discussions were made regarding the effect of PEF technique and the advantage of its combination with other non-thermal methods. Even though there is another similar review on the same topic (reference [1] of the manuscript), the present manuscript is distinctive since the authors focused on discussions regarding the advantage of other approaches in addition to PEF technology.
Nevertheless, the paper needs some improvements to be suitable for publication, therefore major revisions are suggested. Some points should be address prior to a possible publication, specifically:
Major comments:
- Lines 106-115: Please mention Figure 1 somewhere along this paragraph. Please edit Figure 1 aiming to include the terminology/abbreviatures referred in this paragraph, such as AP, DC, etc..
- Table 1: in the column “Pulse width”, please add some note in the lines where no information are available. The authors can use some common abbreviations such as “n.s.” (non-specified) or “n.a.i.” (non-available information).
- Section 3.1, 3.2 and 4: please mention Table 2 along these sections, in proper places of the text.
- Table 1 and 2: it would be interesting to include in Table 1 and 2, an extra column to include the year of publication, aiming to give an idea of how old the study is.
- Section 5 (conclusion): It is missing in this section, a comment/opinion regarding the reason for low (or null) acceptance of the use of these techniques in the commonly used processes in the industry. Please also refer some eventual patents already registered considering these approaches and the name of some companies or countries that are currently using these methodologies.
- As a final comment, I believe that the references’ list and the works cited must be a updated since there are a significant number of works reported along 2021 (and also in 2022), such as the following examples: DOI 10.1016/j.foodchem.2022.132191, 10.3390/foods11081102, 10.1016/j.jfoodeng.2021.110864, 10.3390/foods10112606, etc.. I was checking and only one paper from 2022 was cited, 2 from 2021, and other 2 from 2020 (only reviews), and bigger papers from older years are cited (4 from 2019, 5 from 2018 and 5 from 2017). This is evidence of the need to add some comments from studies reported in more recent years.
Minor comments:
- Line 28: replace “uapplied” by “applied”
- Line 60: replace “by these enzymes” by “by the activity of these enzymes”
- Lines 163, 167, 243: Please mention the name of the authors instead of referencing only the number of the reference as demonstrated for the following example (line 163): “Similarly, Aguiló-Aguayo and colleagues noted higher (…)”
- Line 383: Section 4 must me numbered as section 3.3.
- Table 2: move entry #2 (Apple juice (M. domestica Borkh. cv. Red Fuji)) to the end of the table, to put together all data regarding apple juices (since I believe that authors organized the tables by type of fruit).
- Ref. 12 (line 491): the year of publication is missing.
Author Response
Comments and Suggestions for Authors
Comments to author:
The work described in the present manuscript is consistent with the scope of the journal.
Authors described the utilization of non-thermal pulsed electric field (PEF) as a promising technology to produce juices with improved properties. The review is very well organized by the type of fruit juice and extensively discussions were made regarding the effect of PEF technique and the advantage of its combination with other non-thermal methods. Even though there is another similar review on the same topic (reference [1] of the manuscript), the present manuscript is distinctive since the authors focused on discussions regarding the advantage of other approaches in addition to PEF technology.
Nevertheless, the paper needs some improvements to be suitable for publication, therefore major revisions are suggested. Some points should be address prior to a possible publication, specifically:
Major comments:
- Lines 106-115: Please mention Figure 1 somewhere along this paragraph. Please edit Figure 1 aiming to include the terminology/abbreviatures referred in this paragraph, such as AP, DC, etc..
Reply: The placement of Figure 1 has been done according to the suggestion of the reviewer.
- Table 1: in the column “Pulse width”, please add some note in the lines where no information are available. The authors can use some common abbreviations such as “n.s.” (non-specified) or “n.a.i.” (non-available information).
Reply: Suggestion incorporated in the table (highlighted in green)
- Section 3.1, 3.2 and 4: please mention Table 2 along these sections, in proper places of the text.
Reply: Suggestion incorporated in the table (highlighted in green)
- Table 1 and 2: it would be interesting to include in Table 1 and 2, an extra column to include the year of publication, aiming to give an idea of how old the study is.
Reply: Suggestion incorporated in the table (highlighted in green)
- Section 5 (conclusion): It is missing in this section, a comment/opinion regarding the reason for low (or null) acceptance of the use of these techniques in the commonly used processes in the industry. Please also refer some eventual patents already registered considering these approaches and the name of some companies or countries that are currently using these methodologies.
Reply: Dear reviewer, thank you so much for your valuable suggestion. During the preparation of early drafts of this review. Authors mutually decided to add a section on limitation to use such techniques at industrial level. But adding this section would increase the length of the manuscript. Also, without a proper incorporated section, its not recommended to add anything in conclusion.
- As a final comment, I believe that the references’ list and the works cited must be a updated since there are a significant number of works reported along 2021 (and also in 2022), such as the following examples: DOI 10.1016/j.foodchem.2022.132191, 10.3390/foods11081102, 10.1016/j.jfoodeng.2021.110864, 10.3390/foods10112606, etc.. I was checking and only one paper from 2022 was cited, 2 from 2021, and other 2 from 2020 (only reviews), and bigger papers from older years are cited (4 from 2019, 5 from 2018 and 5 from 2017). This is evidence of the need to add some comments from studies reported in more recent years.
Reply: Citations updated as suggested and highlighted in blue.
Minor comments:
- Line 28: replace “uapplied” by “applied”
Reply: Corrected (Highlighted in green)
- Line 60: replace “by these enzymes” by “by the activity of these enzymes”
Reply: Corrected (Highlighted in green)
- Lines 163, 167, 243: Please mention the name of the authors instead of referencing only the number of the reference as demonstrated for the following example (line 163): “Similarly, Aguiló-Aguayo and colleagues noted higher (…)”
Reply: Corrected (Highlighted in green)
- Line 383: Section 4 must me numbered as section 3.3.
Reply: Corrected (Highlighted in green)
- Table 2: move entry #2 (Apple juice (M. domestica Borkh. cv. Red Fuji)) to the end of the table, to put together all data regarding apple juices (since I believe that authors organized the tables by type of fruit).
Reply: Corrected (Highlighted in green)
- Ref. 12 (line 491): the year of publication is missing.
Reply: Corrected (Highlighted in green)
Dziadek, K.; Kopeć, A.; Dróżdż, T.; Kiełbasa, P.; Ostafin, M.; Bulski, K.; Oziembłowski, M. Effect of Pulsed Electric Field Treatment on Shelf Life and Nutritional Value of Apple Juice. J. Food Sci. Technol. 2019, 56, 1184–1191, doi:10.1007/s13197-019-03581-4.
Reviewer 3 Report
There are some minor suggestions on a manuscript:
1. Is it necessary to state the author name along with the reference number?
e.g. line 151: [34] stated that the higher the hydrogen bonds…..; line 239: In a research by [50], the RA of PPO and POD was reduced; line 243: According to [68]and [67], the denaturation of enzymes …..etc.
2. line 396: "The maximal RA of β-GLUC (118.8%) was achieved at 144.46 Hz.."
Is it means that the treatment enhanced β-GLUC activity of 18% in comparison to the control- (referent value of 100 %).
Author Response
Comments and Suggestions for Authors
There are some minor suggestions on a manuscript:
- Is it necessary to state the author name along with the reference number?
e.g. line 151: [34] stated that the higher the hydrogen bonds…..; line 239: In a research by [50], the RA of PPO and POD was reduced; line 243: According to [68]and [67], the denaturation of enzymes …..etc.
Reply: Accodring to MDPI journal template and requirements, numbering for references in the text is required. But for your reading we have added the references. We shall delete it at further stages. Corrected and highlighted in Pink & Green.
- line 396: "The maximal RA of β-GLUC (118.8%) was achieved at 144.46 Hz.."
Is it means that the treatment enhanced β-GLUC activity of 18% in comparison to the control- (referent value of 100 %).
Reply: There is a typing mistake and we have corrected as: “The maximal RA of β-GLUC was achieved (35 kV/cm for 1700 μs using pulses of 4 μs at 100 Hz in bipolar mode) than heat treatments at different temperatures”.
Round 2
Reviewer 2 Report
The authors have addressed my concerns in the present revision. The modifications and updates performed significantly improved the quality of the manuscript.
I only feel sad because I believe the inclusion of, at least a small, section regarding the limitations to use such techniques at industrial level will be a valuable point for this review. Maybe authors can reconsider this decision before publication. Nevertheless, I consider the paper now suitable for publication.